# From Strategic Narratives to Code-Like Cognitive Models: An LLM-Based Approach in A Sorting Task

**Hanbo Xie[1,2], Huadong Xiong[1,2] & Robert C. Wilson[1,2]** *

[1]School of Psychology, Georgia Institute of Technology, Atlanta, GA, USA
[2]Department of Psychology, University of Arizona, Tucson, AZ, USA

{hanboxie1997,hdx}@gatech.edu, bob.wilson@gatech.edu

## Abstract

One of the goals of Cognitive Science is to understand the cognitive processes underlying human behavior. Traditionally, this goal has been approached by analyzing simple behaviors, such as choices and response times, to try to indirectly infer mental processes. However, a more direct approach is to simply ask people to report their thoughts - for example, by having them Introspect after the fact about the thought processes they used to complete a task. However, the data generated by such verbal reports have been hard to analyze, and whether the reported thoughts are an accurate reflection of the underlying cognitive processes has been difficult to test. Here we take a first stab at addressing these questions by using large language models to analyze verbally reported strategies in a sorting task. In the task, participants sort lists of pictures with unknown orders by pairwise comparison. After completing the task, participants wrote a description of their strategy for completing the task. To test whether these strategy descriptions contained information about people's actual strategies, we compared their choice behavior with their descriptions of the task. First, we compared the descriptions and choices at the level of strategy, finding that people who used similar sorting algorithms (based on their choices) provided similar verbal descriptions (based on the embeddings of these descriptions in the LLM). Next, we generated code based on their strategy descriptions using GPT-4-Turbo and compared the simulated behaviors from the code to their actual choice behavior, showing that the LLM-generated code predicts choice more accurately than chance other, more stringent, controls. Finally, we also compare the simulated behaviors of generated codes with those from standard algorithms and induct the strategies that this code internally represents. In sum, our study offers a novel approach to modeling human cognitive processes by building code-like cognitive models from introspections, shedding light on the intersection of Artificial Intelligence and Cognitive Sciences.

## 1 Introduction

One of the goals of Cognitive Sciences is to understand the cognitive processes underlying human behaviors. Cognitive scientists have long strived for this goal by building computational models that simulate cognitive processes, and have tested these models by comparing their performance to behavior on some task. However, building these models relies heavily on the cognitive scientist's intuition, which is subject to bias and blind spots, while testing these models is limited by the low dimensionality of the behavioral observations, which usually consist of only choices and response times.

---

*Corresponding Author

Instead of this *behaviorist* approach to Cognitive Science, another way to probe human cognition is to simply ask people what they are thinking — that is, to get them to *introspect* about how they are performing the task. Such introspection research has a long and checkered history in psychology going back to the 19th century but despite early interest from the likes of Simon & Ericsson (1984), has had relatively little impact on modern computational cognitive science. A major reason for this lack of impact has been due to the difficulty of handling verbal data, which requires (labor-intensive) transcription and (often subjective) qualitative coding before it can be analyzed, making it hard to be compatible with traditional behavioral analysis and computational modeling. With the advent of Natural Language Processing and Large Language Models (LLMs), there is potential to link introspection in the form of text to computational cognitive models, predicting human behavior and inducting — in a *data-driven* manner — the underlying cognitive processes.

In this paper, we bridge the gap between introspection and cognitive models by using LLMs to generate computer code from verbally reported thoughts. By converting introspection to a series of symbolic operations as represented in a program Lipkin et al. (2023), this approach allows us to generate a bespoke, data-driven cognitive model for each person and test whether the behavior generated by this model is consistent with the behavior the participant actually exhibited.

An ideal paradigm for accommodating such research should have two features. First, it should not be oversimplified so in the task, there are diverse strategies to achieve the goal and the performance depends on the strategy used. Second, it should be highly structured so that the strategies can be easily formed as programming codes for analysis. Christian & Griffiths (2016) wrote a book *Algorithms to Live By*, which introduces a variety of problems that both humans and computers will face, and the algorithms they use to solve these problems. As one of the classical programming problems, sorting tasks is a good fit for our study. Thompson et al. (2022) conducted a large-scale sorting experiment and collected a rich behavioral and strategy descriptions dataset. Therefore, our paper, as an extension of their paper, focuses on this open dataset to investigate how we could generate code-like cognitive models in the sorting task from the strategy descriptions.

Our study leverages the public dataset to develop a pipeline that uses LLM to generate programming codes that could simulate sorting behaviors, based on the strategy description by human participants. We first used text embeddings of strategy descriptions to test their relationship to behaviors and found that the text embeddings are informative to predict strategies discovered from behaviors in the original paper. We then used these descriptions to generate codes that can mimic human sorting behaviors, compared the simulated results with the true behaviors of participants, and found a significantly higher accuracy than the chance level as well as the prior knowledge that LLM can predict. Finally, we inducted those generated codes into known strategies based on their simulated results and found some strategies are highly consistent with those identified from behaviors. In brief, our study shed light on formulating cognitive models from strategy descriptions by generating codes, paving a potential path toward further development of cognitive sciences with LLM.

## 2 Methods

### 2.1 Task and Dataset

We used the dataset from Thompson et al. (2022), which consists of a sorting task in a social learning context. The whole dataset contains 3,408 participants, and each participant needs to complete 13 trials of the sorting task. In each trial, participants sorted six images, which were not previously known to them, into the correct order by choosing any pair to swap positions. Only pairs that were out of order could be successfully swapped (see Figure 1B). When participants think everything is in order, they could click on 'Finish' to end the trial. Then, they will receive feedback about how many comparisons are made and whether the final order is correct or not. The number of comparisons and the outcome will be used to calculate their trial performance. Each participant will need to complete 13 trials of the task, with each task exhibiting random images as well as underlying orders. After they complete

the whole task, they are instructed to write down how they completed the task and what their strategies are, which are used in our analysis.

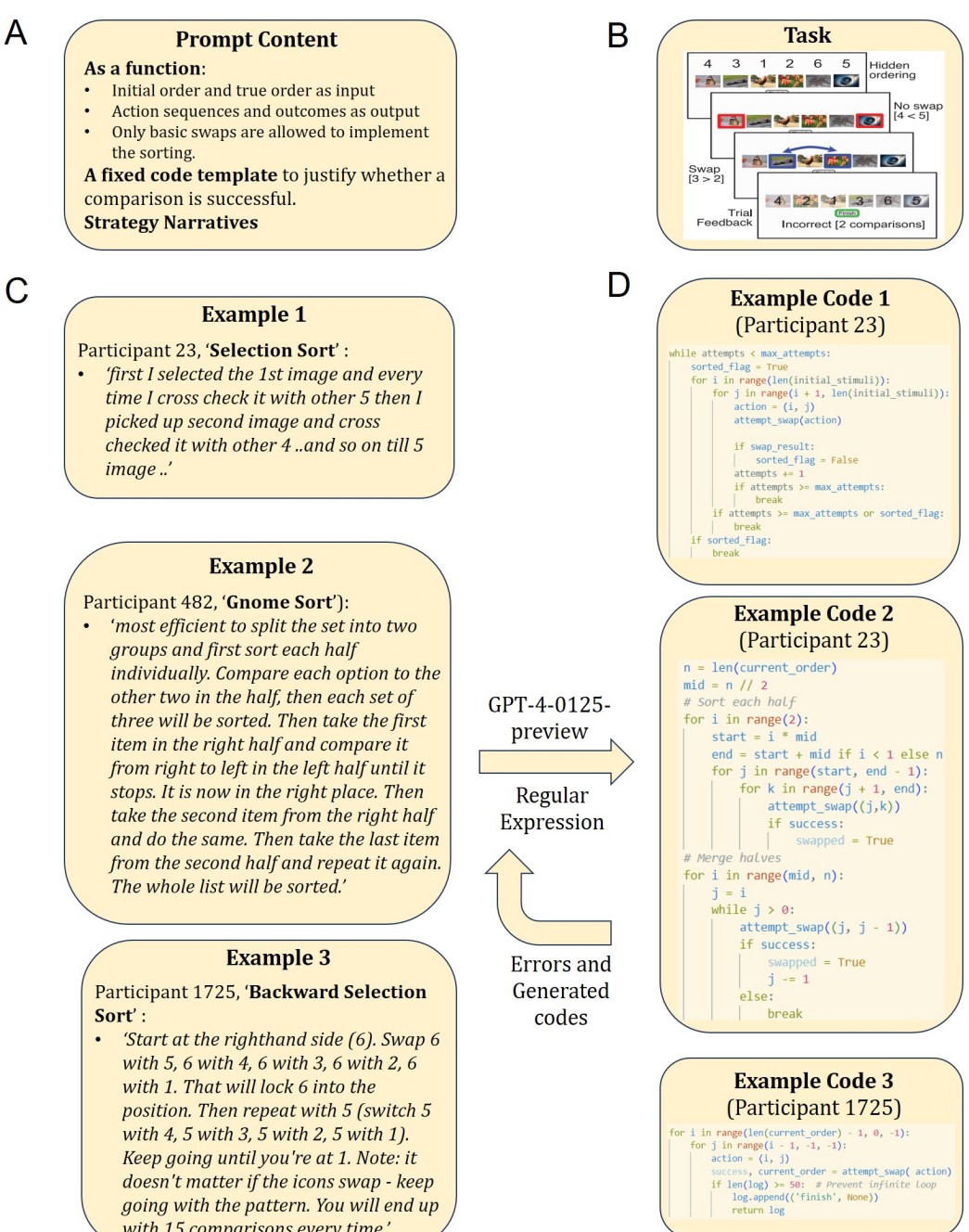

Figure 1: **A**. The Prompt Content to generate codes. **B**. The task procedure. **C**. Example strategy descriptions, with their behaviorally assigned strategies. The participants are best fitted by the codes in their algorithm. **D**. Generated codes of the example participants. For exhibition purposes, we only keep the core logic of the codes.

## 2.2 Generating Executable Codes with GPT-4-Turbo with Zero-Shot Training

We use the most recent version of GPT-4-Turbo, *GPT-4-0125-preview*, to directly generate the codes. To make the codes executable and simulate behaviors efficiently, we restricted the form of the codes in prompts. The principle of restricting the code is to build a human-like sorting function, which could directly take the task input and return sorting behavior, as a general form of cognitive models. Therefore, first, the generated code has to be a function, which accepts the initial order and true order as input. It should produce an action sequence to complete the sorting task. Second, the code cannot use any sorting function, or directly look at the true order to get the sorting behavior. It should simulate how the participants do the task, and use pairwise comparisons. Third, to minimize the variances in justifying the task rule, we directly provide a function template that can tell the 'agent' whether a pair of comparisons is successful or not. Finally, we also insert the participant's strategic description to ask the LLM to try to generate the function aligned with the description (see Figure 1A). The prompts are fixed for each participant but only varying their strategic descriptions.

## 2.3 Behavior Simulations

After all codes were generated and passed the pre-examination for execution, we ran two types of simulations to generate sorting behaviors for different purposes. First, we retrieve the initial order and true order for each participant and each trial from the public dataset and input them into the codes. The code-like models will produce a sorting behavior sequence like what human participants face in the real task. This result was used to evaluate how well the model can predict real human behaviors. We ran the other comprehensive simulation that looped through all possible permutations of the task (for length 6 sequences, there are 720 permutations). We use this result to measure the behavioral outcome of the codes in this task and try to differentiate different strategies indicated by the codes.

To evaluate how accurately the model can predict, for each pair of the predicted and real action sequences, we compare them and count how many actions are predicted correctly. The accuracy of each predicted sequence is computed by the number of correct predictions divided by the length of the true behavior sequence. To evaluate how algorithmically similar each generated code is to each other, we used the comprehensive simulation result and compared each subject to each trial's behavioral sequence. We then averaged each pair of participants' similarity across 720 permuted trials. Specifically, for each pair of behavioral sequences, we define their similarity as the first index the two sequences deviate from each other as the proportion of the longer sequence. In this definition, when two sequences are identical, the similarity is 1. It is formulated as below:

$$S(A, B) = \frac{\text{argmin}_i(a_i \neq b_i)}{\max(|A|, |B|)} \tag{1}$$

# 3 Result

## 3.1 Text Embeddings of Strategic Narratives Reveal Information of Human Sorting Behaviors

Strategic narratives are summaries by the participant after completing the task. The information underlying the strategic narratives should reveal how participants did the task. The original study modeled the participants' sorting behaviors by RNN and inducted 16 commonly used sorting algorithms by model comparisons. Here, we tested how strategy descriptions relate to these algorithms purely inducted from behaviors.

We input the strategy descriptions of all participants into the embedding model, *text-ada-002* OpenAI (2023) and use K-mean Clustering by gathering them into 8 different clusters. We mapped each participant's algorithm discovered from the original study by behavioral data into the cluster. We picked the two most commonly used algorithms (Selection Sort and Gnome Sort) with their forward and backward versions (see the number of participants

identified to all other algorithms by behaviors in table 1) and found out that different algorithms distributed differently in these 8 clusters. More interestingly, the same algorithms but with different implementational directions (i.e., 'from left to right' vs. 'from right to left') exhibit very similar distribution patterns across clusters(Fig. 2). Despite their descriptions may be different semantically, the similar distribution suggests that the text embeddings of strategy description can reveal the underlying algorithmic information. We further tested whether the text embedding is sufficient to classify the two strategies and we found that with PCA to 200 dimensions (with 89% variances explained), text embeddings could well classify the two strategies with SVM with five-fold cross-validation, obtaining an average test accuracy of 88% ± 2%. This result suggests that strategic narratives can indeed reveal algorithms inferred purely from behaviors and could be potentially informative in building cognitive models. We also expand the test to eight common algorithms (merging forward version and backward version of the same algorithm) and 16 algorithms (without merges), and obtain an averaged test accuracy of 37.4% ± 4.6%, 36.3% ± 4.3%, exhibiting a higher performance than chance level.

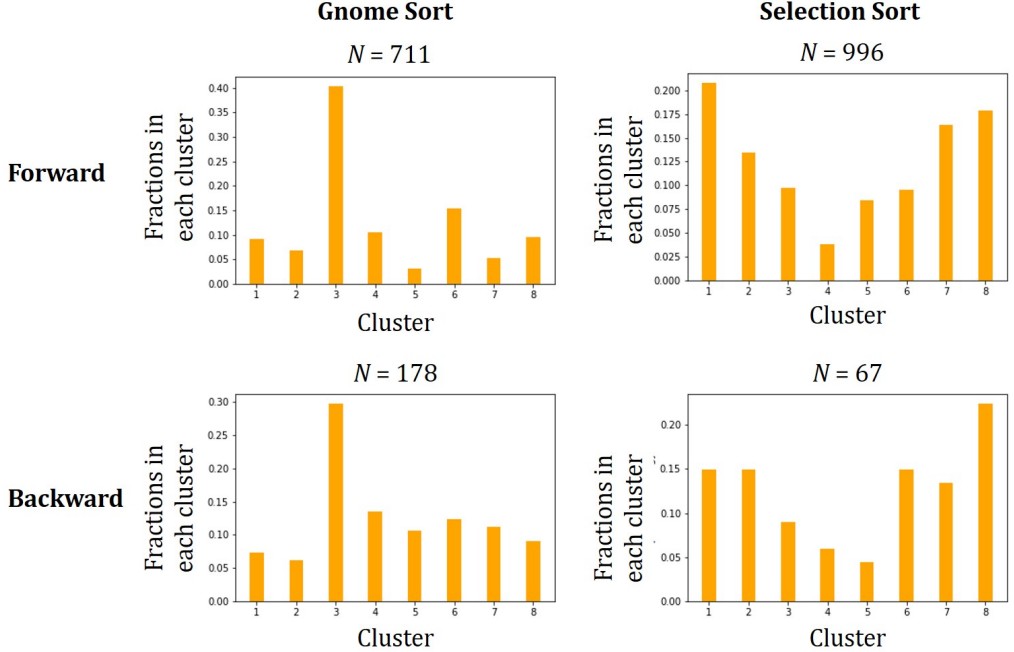

Figure 2: Text Embeddings from different algorithms exhibit different distributions in clusters, and the same algorithms with different versions exhibit similar patterns.

## 3.2 LLM-Generated Codes Surpass Chance and Prior Knowledge in Predicting Human Sorting Behaviors

Since strategy descriptions reveal the behavioral information, we create code-like cognitive models. These models mimic human sorting behavior by prompting GPT-4-0125-Preview with the strategy descriptions. The generated codes can be executable after extracting and debugging from the LLM responses. For each code, we input the task's specific challenges faced by the corresponding participant to predict their sorting behaviors. We compared the true behavior sequence with the simulated behavior sequence at the element level. The result suggests that the model accurately predicts the actions of participants beyond the chance level, with 78% of participants surpassing the chance level (predicting 1 out of 16 possible actions, 15 possible pairs plus 1 'finish' action, see Figure 3A). However, one could argue that the LLM's prior knowledge about sorting algorithms could influence its code generation, enabling it to produce sorting behaviors that align with human actions above chance levels, regardless of the content of the strategic descriptions. Yet, this influence does not imply

random generation nor does it guarantee human-like behavior outcomes. To confirm the LLM makes use of the strategic descriptions to generate codes, we ran a simplified permutation test in which we permute code-participant correspondence to evaluate a null level of predicting human sorting behaviors by randomly generating sorting codes. Our results show that if the LLM does not use strategic descriptions to generate the codes, the mean prediction accuracy for behaviors is 14.8% (95% CI: [14.3, 15.3]). Therefore, the true mean accuracy is higher and out of the null distribution from purely prior knowledge, with 49% of participants surpassing the prior knowledge. This result suggests that our model performance surpasses the chance and prior knowledge level, indicating its robust performance in predicting human sorting behaviors as cognitive models.

We further take a look at how the prediction accuracy could vary across trials and participants since we suppose algorithmic information revealed by strategic descriptions is different in these conditions. We first tested whether as the trial went on, the prediction accuracy of the model increased accordingly. The result shows a strong correlation between the trial number and the predicting accuracy (see Figure 3B, $r = 0.82$, $p < 0.001$). This indicates a learning effect in their task that a better performance occurs in later trials as already indicated by the original research. It might also be because the strategy descriptions happen at the end of the task, and it is more likely for participants to summarize their thoughts from the most recent trial (last trial).

We also test whether the prediction accuracy varies across participants as their task performance varies. We first ran an overall correlation across all participants, finding that task performance is significantly positively correlated with predicting accuracy ($r = 0.21$, $p < 0.001$). This may indicate that participants with better performance may have clearer expressions about their strategy, and their behaviors are less random which could be easier to be fitted by sorting codes. However, as we observed the plot and found there is a mixture of distributions across these two variables (see Figure A1), we plot and calculate the correlation by previously behaviorally assigned algorithms. We still picked the two most commonly used algorithms here to calculate their correlations. As shown in Figure 3C and 3D, both algorithms of participants exhibited significant correlations (Gnome sort: $r = 0.19$, $p < 0.001$; Selection sort: $r = 0.31$, $p < 0.001$). However, their distributions of both task performance and prediction accuracy are quite different, suggesting that either the LLM may have inducive biases into some particular sorting algorithms or descriptions from some algorithms are more explicit and comprehensible, but those from others are more implicit and challenging to convey. This variation in semantic transparency among strategies could significantly influence the effectiveness of the LLM's code generation and subsequent behavior prediction.

In sum, the generated codes exhibit a robust prediction accuracy above chance and prior knowledge and reveal individual differences and temporal relationships from participants' behaviors, suggesting their potential to serve as predictive and explainable cognitive models.

### 3.3 Comprehensive Simulated Behaviors in the Task Induct Strategies Underlying LLM-Generated Codes

Since these codes can predict the behaviors, and align with the simulated behaviors and strategy descriptions, we are interested in the underlying cognitive processes they represent. In sorting tasks, the most important cognitive process is the strategy or algorithm humans use to sort unordered lists. As revealed in the original study, researchers used RNNs to identify 16 standard and commonly used sorting algorithms for most participants. Following their hypothesis, we evaluate the similarity between each generated code and hypothetical algorithms. By running comprehensive simulations with the standard algorithm codes, we compare the simulated behaviors of each generated code to those of the standard algorithms. When a generated code's simulated behaviors exactly match those from a standard algorithm (*similarity* = 1), we attribute the code to that algorithm. We compared our results with the original study, which based its strategy attributions on behaviors. Attributions between codes and behaviors were consistent for 28.2% of participants.

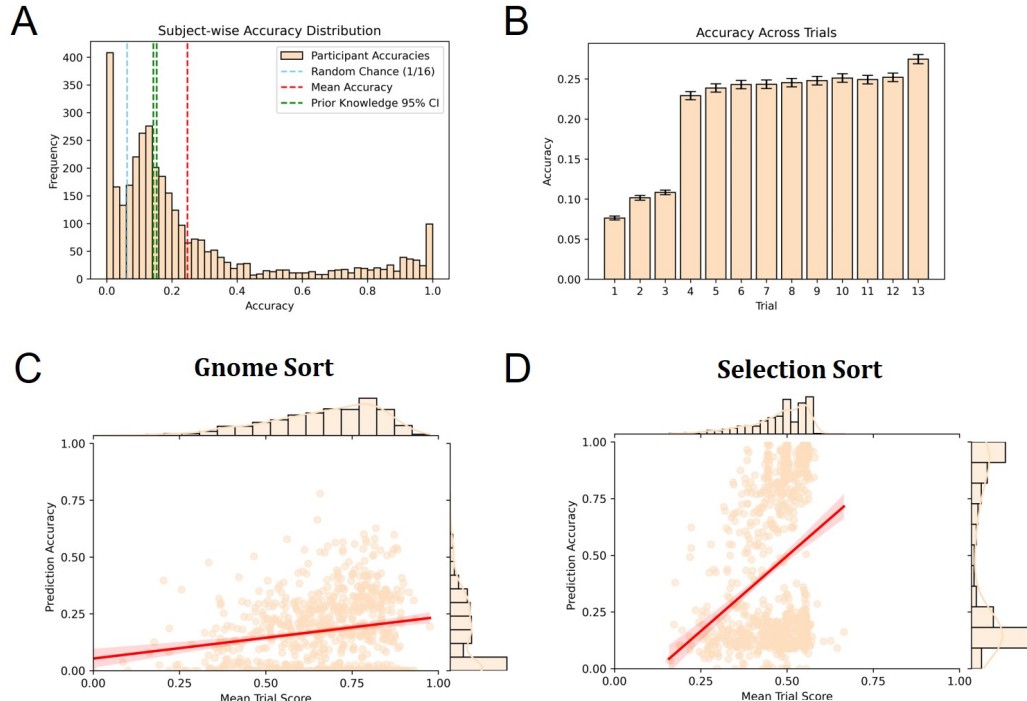

Figure 3: **A**. The prediction accuracy of generated codes surpass random chance and prior knowledge by the LLM. **B**. The generated codes capture learning effects in human behaviors. **C**, **D**. The generated codes can capture individual differences in the task. For different algorithms, the LLM has different performance and variability in capturing such individual differences.

Moreover, we used a confusion matrix to compare LLM-inferred strategies with behaviorally assigned strategies, examining the alignment and discrepancies between LLM-generated computational models and algorithms based on actual participant behaviors. The result shows that several strategies are well aligned (e.g., Selection Sort, Successive Sweeps) but also indicates the shortcomings of LLM in identifying some other strategies (e.g., Gnome Sort and Backward Gnome Sort) from purely strategy descriptions. To confirm whether this shortcoming in identifying strategies is because the LLM has some inductive biases to some particular algorithms or people from some of the algorithms find it hard to describe the algorithm clearly, we ran a recovery test that we instruct GPT-4-0125-Preview to describe each algorithm in this task context, and ran a completely same pipeline as above. We found that even for the LLM-generated descriptions, the LLM cannot correctly identify those biased strategies (e.g., Gnome Sort, see Figure A2). The recovered confusion matrix exhibits a very similar pattern to Figure 4, which suggests there is a systematic inductive bias in sorting algorithms in the LLM.

## 4 Discussion

### 4.1 Introspections, cognitive processes, and behaviors.

Building precisely predictive and interpretable computational models is crucial to the field of cognitive sciences. Introspections can be informative in probing thought processes rather than purely inferring from behavioral observations. In our study, we showed that text embeddings of strategy descriptions can well predict strategies discovered from behavioral datasets. We are not the first ones to observe such effects. Previous research revealed that both self-report ratings and Think-Aloud as natural languages can be used to predict human choice behaviors, as well as probe the underlying cognitive processes with computational

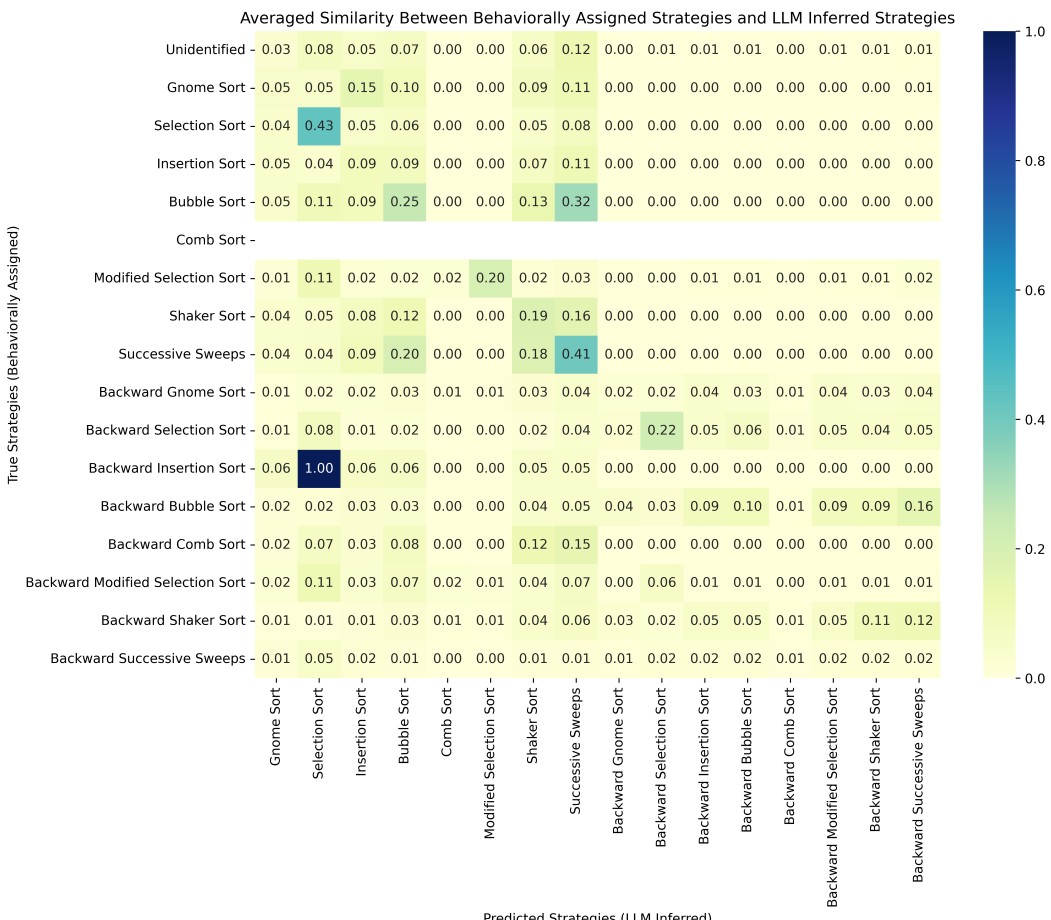

Figure 4: The confusion matrix compares the behavioral similarity of code-generated simulations to standard algorithms, organized by behaviorally identified strategies. It quantifies how closely codes, derived from participants' strategy-based descriptions, align with algorithmic benchmarks in behavioral outcomes.

models and neural networks (Morris et al., 2023; Xie et al., 2023). Our study again highlights the importance of introspection in building computational cognitive models of human behaviors.

## 4.2 Code-like models and behaviors.

In our study, we generated codes with the LLM based on the strategy descriptions from human participants, which are later used to predict behaviors and induct strategies that they use throughout the task. With the code-like models achieving higher accuracy than chance level and surpassing the prior knowledge the LLM can produce, the descriptions do contain information that the LLM could utilize. However, the current model performance is still inferior to the purely behavioral RNN model performance Correa et al. (2024); Ellis (2024), indicating that zero-shot learning is not yet sufficient to comprehend behaviors fully. The generated code is mostly deterministic and optimal, overlooking human-centric cognitive features like memory, action biases, and randomness. To further encapsulate human-centered behavioral features, program induction could be utilized, treating the generated code as a prior.

### 4.3 LLM, Introspections, Code-like Models, and Behaviors.

Interestingly, we also found that the LLM exhibits strong inductive biases towards some sorting algorithms. This bias affects both strategy attribution and model performance for specific datasets. The relationship between the LLM, introspections, code-like models, and behaviors can be regarded as a hierarchical Bayesian structure, where the domain knowledge of LLM serves as the prior when generating codes conditioned on the introspections. Therefore, correcting inductive biases of the LLM is as crucial as fitting the codes to behavioral data when generating precise and interpretable code-like models. Ellis (2024) revealed that a tuned-prior LLM could outperform a pre-trained-prior LLM in generating codes that fit behaviors. This suggests the importance of teaching the LLM to be a good 'scientist' in proposing models (generating codes), rather than serving as a mere cognitive model to predict behaviors.

### 4.4 Limitations and Future Directions.

**Introspections and Task Settings.** In the original study, the strategy descriptions are mainly used for communication purposes and are passed on to the next generation of participants to learn. This means participants may not fully reveal their behavior, instead telling what they think is optimal in the task. Additionally, each participant only provides one paragraph of introspection at the end of the task, which overlooks the dynamics of their behavioral evolution. A higher temporal resolution for probing thoughts, such as at each trial, would better capture behavioral patterns. The lack of demographic information also prevents us from investigating individual differences in task performance and strategy descriptions. These task settings constrain the predictive performance of code-like models, which could be better designed for a separate task.

**The Biases of Large Language Models.** This language-based program induction task requires LLMs to have the integrative capability to understand the instructions and strategy descriptions and generate codes correctly. We attempted to use 'code-LLaMA-34b-Instruct-hf' for the task but found the model failed to generate executable codes for all participants without human intervention, preventing us from further comparing their actual performance. We also used the most recent open-source models, 'LLaMA3-8b' and 'LLaMA3-70b', to classify the strategies participants used based on their descriptions. We found our currently used model, GPT-4-0125-preview, exhibits better performance than the other two models (Fig. A3). Despite the superior performance, the current model still exhibits obvious inductive biases toward some algorithms, hindering the model from correctly generating codes for specific groups of participants. One easy way to investigate whether different LLMs exhibit different biases is to use a mixture of models to optimize general performance. More importantly, correcting model biases by fine-tuning or instruction-tuning can help 'teach' models to distinguish different algorithms, benefiting downstream program induction tasks.

**Interpretability of Generated Codes.** In this paper, we have shown that the generated codes can serve as a form of cognitive model that can predict human sorting behaviors based on their strategy descriptions. However, how the codes represent the cognitive processes is poorly understood. In our analysis, we compare each participant's generated codes to standard algorithms, trying to see how language-based program induction captures human behavior compared to purely behavior-driven models. How the strategy descriptions provide information beyond behavioral observations and the standard algorithm hypothesis space is yet to be investigated. One future avenue could be decomposing the codes into primitives (e.g., operations, index, loop) and instructing LLMs to build purely on these primitives. By analyzing how each generated code organizes these primitives, we can understand how codes vary from each other individually, and potentially identify whether biases stem from humans or LLMs among these primitives.

Author Contributions

H.X., H.X., and R.W. conceptualized the project. H.X. and H.X. designed the analysis pipeline. H.X. implemented the analysis and drafted the manuscript. H.X., H.X., and R.W. reviewed and revised the manuscript for submission.

Acknowledgments

We thank Thomas. L. Griffiths, Bill Thompson, and Ted Summers for helpful discussions and providing the datasets for our analysis. This work was funded by a SCIALOG Award #29079 from the Research Corporation for Scientific Advancement (to RCW) and the OpenAI Researcher Access Program (to HX). We thank the four anonymous reviewers and area chairs for their constructive feedback. The authors declared no competing interests.

Data & Code Availability

The dataset and the codes for this paper can be found at https://github.com/xhb120633/code_like_cog_model.

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

# A Appendix

## A.1 Prompts of Generating Codes from Strategic Narratives

For each participant, we extract their strategic descriptions and insert it into the prompt. The complete prompt is like below:

*Create a Python sorting function 'human_like_sorting',which is directly executable and effective for a general sorting task, based on a strategic approach to sorting a list of items. The function should simulate the decision-making process humans might use when sorting, without direct comparison to a known 'true order'. The function must:*

1. *Accept 'initial_stimuli' as a list of items to be sorted, with 'true_order' available only to the internal 'attempt_swap' function for providing feedback on swap actions. 'True_order' represents the correct order of items and is used to simulate feedback on whether a swap action brings items closer to their desired order:*

```python
def attempt_swap(current_order, true_order, action):
    index1, index2 = action
    if 0 <= index1 < len(current_order) and 0 <= index2 < len(current_order)
        and index1 != index2:
      item1, item2 = current_order[index1], current_order[index2]
      true_pos1, true_pos2 = true_order.index(item1), true_order.index(item2)
      if (true_pos1 > true_pos2 and index1 < index2) or
          (true_pos1 < true_pos2 and index1 > index2):
        current_order[index1], current_order[index2] =
        current_order[index2], current_order[index1]
        return True, current_order
    return False, current_order
```

2. *Return a log of actions attempted, regardless of the outcome, formatted as tuples of item indices attempted for swapping, as well as the outcome of this attempt. A 'finish' action signals the end of the sorting process. This log tracks the sorting actions as well as the outcome of the comparison, simulating a step-by-step decision-making process.*

3. *Contain all necessary logic for sorting within the function, making it capable of sorting lists of any length using basic operations. The function should not depend on external libraries or Python's built-in sorting functions for its core sorting logic. To avoid infinite loop, the function should set a hard limit up to 50 comparison attempts and always ended up with a 'finish'. All the basic functions should be correctly used (if used).*

4. *Refrain from using 'true_order' for direct comparisons with 'current_order' to assess sorting completion. Instead, infer the completion through the sorting process itself, akin to how a person might determine they have finished sorting without knowing the exact 'true_order'. 'true_order' can only be used in the 'attempt_swap' function.*

*Focus on generating executable, valid Python code that effectively sorts 'initial_stimuli' into order. The code should first and foremost be functional and capable of performing the sorting task well under general conditions. Once functional validity is ensured, align the code as closely as possible with the strategic description provided, within the bounds of simulating human-like sorting behavior under fair computational constraints.*

*Please provide only the implemented function, ready for direct execution with 'initial_stimuli' and 'true_order' inputs. Don't include any comments, explanations, notes, or examples.*

*The strategy description is {strategy_description}.*

For best reproducibility, we always fixed the hyper-parameters, with max_token = 1024, temperature = 0.1, seed = 2024, with all other hyper-parameters set to their default values.

## A.2 Representational Similarity Analysis: Aligning Generated Codes, Descriptions, and Simulated Behaviors

To investigate how well the strategic descriptions, generated codes, and simulated behaviors are aligned, we use inter-subject RSA to validate their correlations. The very first step of RSA is to define a similarity matrix for each level of datasets. For strategy descriptions, we obtain the text embeddings by inputting the text into the *text-ada-002* model by OpenAI (2023). We compute cosine similarity for each pair of participants to obtain the similarity matrix. For the generated codes, we use CodeBERT to obtain the embeddings of codes and similarly compute the inter-subject similarity matrix of code embeddings. For the behavioral similarity matrix, we computed behavioral similarity in the same way as described in the methods section (2.3).

After three inter-subject similarity matrices were computed, we correlated the upper triangular, non-diagonal elements of flattened similarity matrices to compare representational structures without redundancy or self-comparisons. To test the correlation, we used the Spearman correlation and permutation test with 2,000 permutations to define whether the correlation is statistically different from the null distribution.

Our results suggest that all three pairs of correlations are out of null distributions (description-code correlation: $r = -0.07$, $95\%CI = [-0.0008, 0.0008]$; description-simulated behavioral correlation: $r = -0.01$, $95\%CI = [-0.0008, 0.0008]$; code-simulated behavior correlation: $r = 0.18$, $95\%CI = [-0.0008, 0.0008]$), indicating significant strong alignments among strategic descriptions, generated codes, and simulated behavior.

### A.3 Behaviorally Assigned Strategies Sample Summary

| Algorithm | Regular | Backwards |
|---|---|---|
| Selection sort | 996 | 67 |
| Gnome sort | 711 | 178 |
| Shaker sort | 71 | 23 |
| Modified selection sort | 63 | 58 |
| Bubble sort | 47 | 38 |
| Successive sweeps | 44 | 14 |
| Insertion sort | 19 | 1 |
| Comb sort | 0 | 8 |
| Unlabeled | 1112 | - |

Table 1: Algorithm Samples Discovered from Behaviors. Source from Thompson et al. (2022)

### A.4 Overall Subject-wise Task Performance and Prediction Accuracy

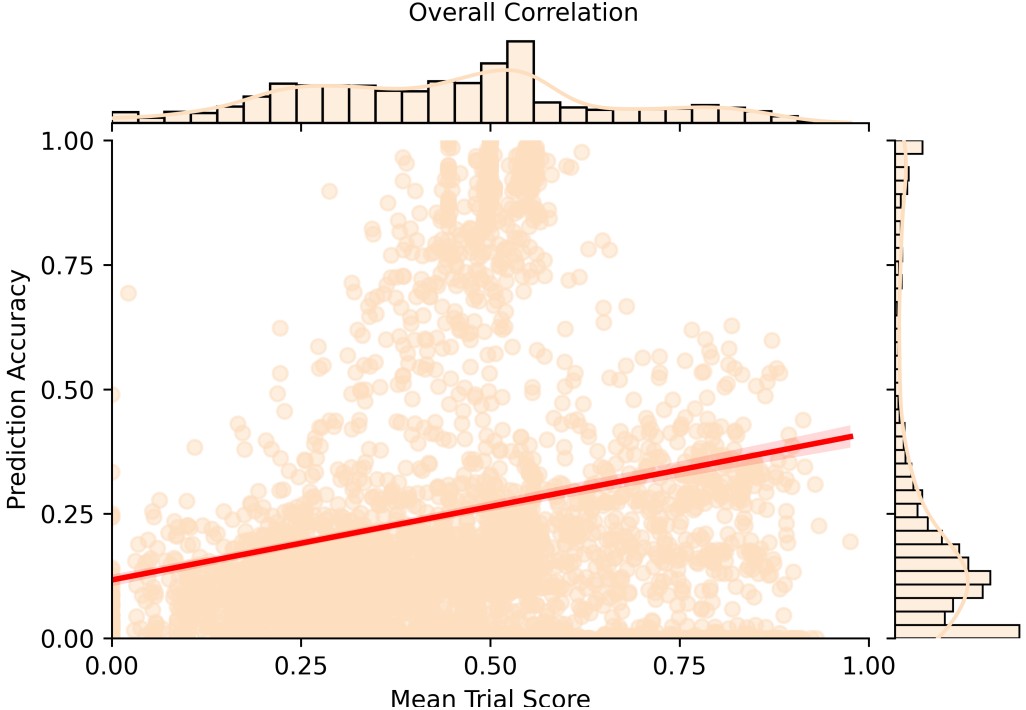

Figure A1: The overall correlation between individual task performance and prediction accuracy. The distributions for both task performance and prediction accuracy suggest a mixture of distributions in the dataset and it is necessary to isolate the algorithms.

## A.5 Recovery Test in LLM for Code-Generations

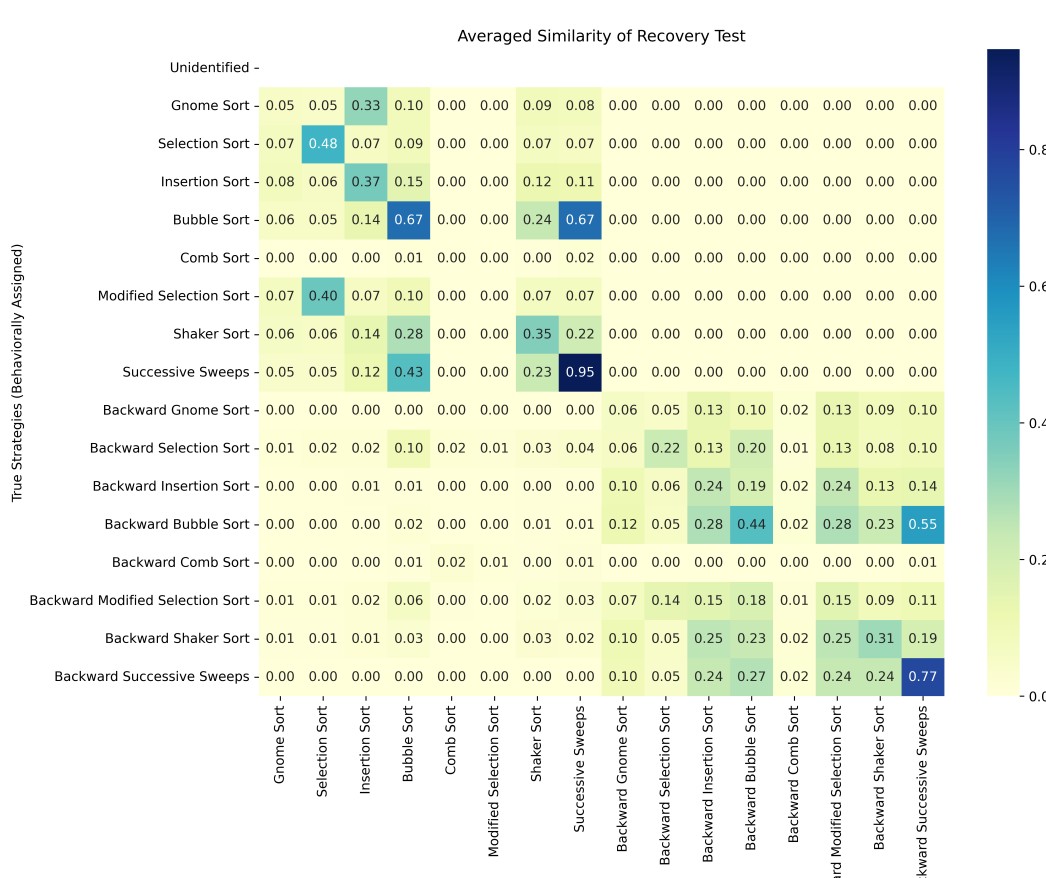

Figure A2: Confusion matrix in recovery test. This recovery test indicates biases towards algorithms similar to those on human data, such as Gnome Sort and Backward Gnome Sort.

## A.6 Strategy Classifications of Several LLMs

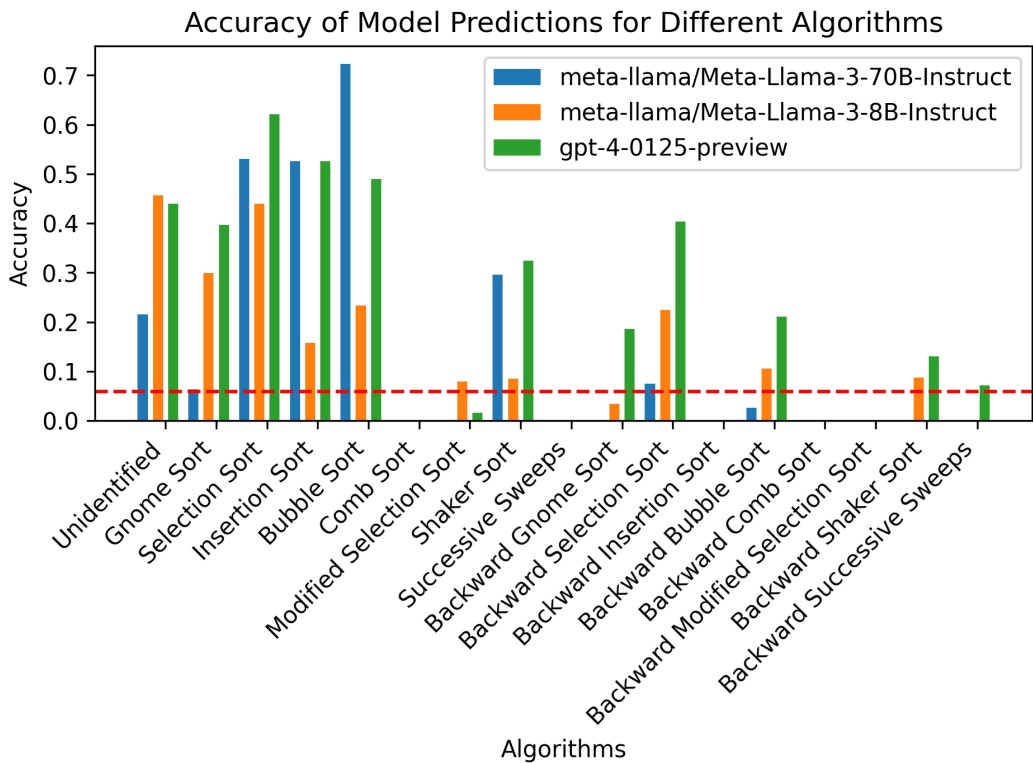

Figure A3: GPT-4-0125-preview generally exhibits better performance than the two most recent versions of LLaMA.

