# OpenReview forum: "From Strategic Narratives to Code-Like Cognitive Models: An LLM-Based Approach in A Sorting Task"
_colmweb.org/COLM/2024/Conference — COLM_

### Official Review · Reviewer_6wRT · 2024-05-03

**Rating:** 6
**Confidence:** 5
**Ethics Flag:** 1

**Summary:**

This paper highlights the challenge of analyzing introspective free-response data in cogntiive science, and proposes to use LLMs to generate code implementing the strategies people report. It shows that this technique successfully generates code that approximates some aspects of the human behavior, and outperforms various control conditions. Overall, I thought the approach was somewhat novel and interesting. However, what I felt was missing was a sense of what cognitive insight was gained by converting the free responses into code, and more discussion of the limitations and drawbacks of the approach.

**Reasons To Accept:**

* This is an interesting approach to analyzing human free-response data.
* The results show that the code generated from the free responses is predictive of their behavior.
* There is some interesting analysis of the relationship between subject performance and the accuracy of the code predictions, as well as learning effects.

**Reasons To Reject:**

* My main concern about this paper is that it's not clear what, cognitively, we learn by converting free responses into code. The paper could be more compelling if it demonstrated some insight that the analysis yields for cognitive science. Just predicting behavior above chance is not sufficient; indeed, as the authors note, an RNN-based approach would predict the behavior better (presumably because it can also account for biases etc.) The authors seem to hint that assigning participants to algorithm buckets is the cognitive insight ("In sorting tasks, the most important cognitive process is the strategy or algorithm humans use to sort unordered lists."), but it is not obvious to me how much we learn from this (especially since it seems fairly weakly correlated with the behavioral sorting, and shows notable biases as the paper notes).
    - A very basic analysis to start would be to see whether the generated code predicts participant behaviors better than the closest ground-truth algorithm to that code, for example. This is similar to the Fig. 4 analysis, but again, with more of an eye to explaining what the participants actually did and how that links to the self report, rather than something about the code/algorithm itself.
   - It would be much more compelling if the code-based analyses brought some new kind of insight that could not have been gained from behavior alone, though I am not sure what exactly that might be.

* More fundamentally, the idea of converting the hypothesis space for cognitive experience from the rich flexibility of natural language or behavior into code requires abstracting away many details that might be important to understanding the complexity of human cognition. If we think that code is only an approximation to the underlying messy way cognition actually works (cf. https://www.cell.com/trends/cognitive-sciences/fulltext/S1364-6613(10)00124-5) then we may be missing important details by moving to that level of abstraction. For example, there may be systematic biases, order effects, etc. in the human behavior that don't emerge in their self-report (many cognitive biases don't), but that nevertheless drive key aspects of their behavior. When we analyze humans through code that approximates their behavior, however, we abstract away those biases and effectively treat them as noise in the data. We also tend to restrict ourselves to research tasks (like sorting) that are amenable to code like analyses, rather than understanding rich naturalistic human behavior. Of course, research and analyses often require such simplifying abstractions, so there is certainly potential utility in the method. But I felt that a paper proposing a new method of analysis, that might be adopted by other researchers, should be more explicit about these tradeoffs in its assumptions.

---

> ### Author Rebuttal · Authors · 2024-05-30
>
> We thank the reviewer for the insightful observations and valuable feedback on our work. We share many of your concerns about this work.
>
> **Comparing generated code with ground-truth algorithms:** We compared participant behavior predictions generated by GPT-4 with ground-truth algorithms from RNNs. The generated codes performed worse (mean accuracy 30.5% vs. 40.4%), but GPT-4 performed better for more subjects. This suggests that GPT-4-generated codes may worsen predictions or that RNN algorithms capture some biases more accurately.
>
> We used GPT-4 and LLaMA-3-70b to attribute a standard algorithm from 16 given algorithms, then used this prediction as standard approximated codes. Changing models or tasks did not improve accuracy compared to the original GPT-4 codes, directing the idea that RNN-attributed algorithms capture biases better.
>
> Data inspection showed participants better predicted by RNNs primarily used the 'selection sort' strategy, involving simple heuristic steps. GPT-4 was sometimes misled by subjects’ descriptions, causing errors. For some, descriptions didn't match actions, leading to incorrect code generation. In other cases, GPT-4 implemented 'selection sort' more efficiently, causing errors. Simple heuristic descriptions, like 'swap each pair of images', also confused GPT-4.
>
> These findings indicate our prompts need refinement. We'll use advanced prompting techniques to better guide LLM agents in solving this task.
>
> **Fundamental utility and limitations:** We acknowledge that converting cognitive experiences from natural language or behavior into code requires significant abstraction. Currently, we don't expect LLMs to fully automate scientific discovery but to aid in generating hypotheses through code-like cognitive models. Traditional cognitive modeling involves iteratively fitting models to data, which can lead to overfitting and may miss important processes. LLM-generated codes expand the hypothesis space for cognitive models, serving as an initial step and benchmark. While LLMs can't capture all biases, they offer a human-understandable medium that aids model-building. This approach allows us to leverage LLMs' knowledge of sorting algorithms and human language to gain insights into human problem-solving strategies.

---

> > ### Comment · Reviewer_6wRT · 2024-06-04
> >
> > Thanks to the authors for their response; my understanding is that the authors broadly agree with the issues raised, and plan to address them in the future. Thus, I maintain my prior assessment of the paper.
> >
> > I do want to comment on one aspect of the response regarding the utility. The authors note that they hope their method will "aid in generating hypotheses through code-like cognitive models [by] they offer[ing] a human-understandable medium that aids model-building [...] to gain insights into human problem-solving strategies." However, in cognitive science participant introspection has often served this purpose without the need for language models to translate it into code; if a scientist will need to interpret the results after anyway, translating into code will not necessarily help them, especially (again) if that translation is biased and lossy. Thus, again, I think that the utility would ideally be motivated, by demonstrating it enables some clearly novel insight into the data.

---

> > > ### Author Response · Authors · 2024-06-07
> > >
> > > Thank you very much for your insightful feedback.
> > >
> > > We appreciate the reviewer's recognition of the potential value of introspections in providing richer data compared to pure behavioral data. Indeed, our approach aims to leverage the hypothesis-free nature of introspections to gain deeper insights into human cognition. We acknowledge that there are various methodologies beyond generating codes from introspections using LLMs that can illuminate the cognitive processes underlying these introspections. In parallel with the current study, we are also exploring other avenues to analyze introspections with LLMs at different levels.
> > >
> > > We believe that generating codes through LLMs offers a promising avenue of investigation, as it allows for the direct construction of individual cognitive models without relying solely on researchers' interpretations. These models are designed to not only mimic participants' behavior in a task but also to represent their general thinking patterns throughout the task. As a result, they have the potential to generalize and predict out-of-distribution behavioral data. In our study, the codes generated by the LLMs are explicitly instructed to adapt to sorting tasks of varying lengths, capturing the underlying algorithmic patterns rather than merely replicating behaviors.
> > >
> > > We acknowledge that the current performance of our approach is not yet optimal and that several biases, including those inherent to LLMs, may be present. Nevertheless, we are confident that by optimizing the codes through program inductions from behaviors or refining our prompts and LLMs, we can achieve more robust performance. This would enable us not only to compare the generated codes to standard algorithms for attribution but also to conduct specific analyses, such as code clustering, to uncover additional insights.
> > >
> > > Thank you once again for your valuable feedback. We look forward to further improving our approach and exploring the rich potential of introspections in cognitive research.

---

### Official Review · Reviewer_ZEwv · 2024-05-11

**Rating:** 4
**Confidence:** 3
**Ethics Flag:** 1

**Summary:**

This paper aims to show that there is a parallel between humans and LLMs in solving problems which are solvable with specific sorting algorithms.

**Questions To Authors:**

Are you giving the name of the algorithm in the prompt?

**Reasons To Accept:**

The paper investigates the parallel between humans and LLMs.

**Reasons To Reject:**

Notably, LLMs are good memorization devices and these devices have been trained on code generated by humans. Then, these machines are really able to build up code of notorious sorting algorithms. The purpose of this study is then really unclear.

---

> ### Author Rebuttal · Authors · 2024-05-26
>
> Thank you for your review. We appreciate your comments and have addressed the main concerns below.
>
> 1. **Purpose of the Study**
>    The reviewer questioned the utility of using LLMs in our research. Our study investigates not the capability of humans and LLMs to solve the sorting problem, but their potential to reconstruct human cognitive processes in algorithm development based on human descriptions in naturalistic languages. This approach allows us to explore how LLMs leverage their inherent knowledge of sorting algorithms and humans’ language to simulate human problem-solving strategies, offering a unique intersection between AI capabilities and cognitive science insights.
>
> 2. **Contributions to Cognitive Science**
>    The contribution of our research lies in demonstrating how LLMs can be utilized to model and understand human algorithmic thinking. By analyzing how LLMs reconstruct full algorithms from human descriptions, we gain insights into potential cognitive processes employed by humans, thus providing a novel method to explore human cognitive strategies through AI on a more detailed level.
>
> 3. **Response to Specific Question**
>    We do not specify algorithm names; these algorithms are clustered in the previous study. In generations of codes, only human participants’ strategy descriptions are provided.

---

### Official Review · Reviewer_fG11 · 2024-05-11

**Rating:** 6
**Confidence:** 3
**Ethics Flag:** 1

**Summary:**

This paper uses LLM to translate participants' written strategy into program code and see if the code simulates their actual behavior. Usually, such translation is done by human labor. LLM can reduce costs and enable larger-scale cognitive experiments.

**Reasons To Accept:**

The purpose of this paper is to establish a technical foundation/verification for a tool to be used in future large-scale cognitive science research. This is a really valid purpose, and an AI/technical conference like COLM is a good place for papers like this. The motivation is clear. The paper is well-written. And the analysis seems detailed.

The authors found that vanilla LLMs cannot surpass an RNN at the moment.

**Reasons To Reject:**

The paper could have explored more LLM strategies/techniques to see whether the gap between LLM and RNN can be closed with fine-tuning or prompt engineering. If not, then it would be nicer to know how other researchers in the field can hope to close the gap. Also, LLM's ability lies with generalization -- a behavioral RNN trained on one task might not generalize to another task. LLM however can. It might be nice to evaluate more than one dataset/task.

I do think this is an interesting paper. I don't know where the bar of COLM should be (since this is the first year). If this paper is submitted to the CogSci conference, it will vote for acceptance. I wish the paper had more technical contributions or explored more on the methodology front.

---

> ### Author Rebuttal · Authors · 2024-05-30
>
> We thank the reviewer for the positive feedback on our paper. We also hope COLM could be a good place for this paper.
>
> **Addressing the gap between LLM and RNN:** One approach to close the gap between LLM and RNN performance is to consider biases and random deviations from the described strategies. While RNNs easily handle these effects, the LLM-to-algorithm approach tends to ignore them unless explicitly described. A potential solution is for the LLM to output algorithms that generate likelihoods instead of hard choices. This would allow for a principled combination with other biases and suboptimalities inferred from choices. For example, randomness could be accommodated with a softmax parameter.
>
> **Prompt Engineering and Inductive Bias:** We recognize that different prompts can introduce various inductive biases. This is a valuable consideration for future work to potentially close the gap between LLM and RNN performance.
>
> **Regarding the dataset and experimental design:** It is important to note that our initial work used a preexisting dataset from an experiment not designed for this type of analysis. The original study ignored the verbal descriptions, leading to a limited number of trials per subject and after-the-fact strategy summaries. Moving forward, acquiring verbal data in real-time as participants complete the task, for example, via Think Aloud procedures, and including more trials and tasks per subject would enable more detailed analysis and better application of our approach.

---

> > ### Comment · Reviewer_fG11 · 2024-06-04
> >
> > I acknowledge I have read the response.

---

### Official Review · Reviewer_myFw · 2024-05-12

**Rating:** 7
**Confidence:** 3
**Ethics Flag:** 1

**Summary:**

The paper proposes to use people's post-collected unstructured descriptions of their behavior in a human study (sorting task) to generate a computer code using LLMs. This code can be then run as a simulation and compared to the actual human behavior shown in the task requested, helping to shed light on the accuracy of the post-explanatory self reporting of the participants. Using GPT-4, the attributions between code and behaviors were consistent in 28% of the cases on a study of about 3k participant samples. Inductive biases of GPT-4 towards some more common sorts are shown and discussed.

**Questions To Authors:**

- from the paper it sounds like the LLM generating code has some difficulties with positional statements ('left to right'/'right to left') -this could be a broader fundamental issues worth further exploring?

- the prediction performance correlated with task accuracy - who were the participant? Are the higher performing ones more likely to have computer science education and with that default to sorting algorithm description that is more comprehensive for LLM to generate code?
If so, how does this generalize to other tasks - the low-performer case where the candidates try to describe their more "ad hoc", unstructured behavior can be of more interest in many studies, but sounds like LLMs would not be applicable for that?

**Reasons To Accept:**

The paper shows quite an innovative quantitative way to evaluate participant testimonies in behavioral studies using LLMs. It is well written and easy to follow.

**Reasons To Reject:**

- Only one model and one test task is used in the study - it would be interesting to see how a range of different LLMs performs in this tasks (in relation to the found inductive biases for example; i would assume this can be a matter of LLM training data; with some luck one could find a few open-source LLMs that vary in this respect).
- i'm not entirely convinced the full strategy description is relevant for the code generation; the authors show that the generation is significantly better than without it, but it would be interesting to check for inductive biases on a small set of keywords, for example if saying something like "then I select" can already trigger a full selection sort implementation without further ado.

---

> ### Author Rebuttal · Authors · 2024-05-30
>
> We thank the reviewer for the positive feedback on our paper.
>
> **Regarding the use of different LLMs:** Due to limited time and resources, we employed LLaMA-3-70b to address a similar task. We clustered human algorithms based on their descriptions and utilized the standard algorithm clustered by the LLM to predict human behavior. The results were comparably good (mean accuracy: 23.6%). The predictions made by different LLMs were consistent for most individuals (correlation = 0.55). Our findings indicate that using both GPT-4 and LLaMA-3-70b does not significantly improve accuracy, as over 75% of participants' accuracy differences were within 5%.
>
>
> **On the suggestion to try keywords and key phrases:** This is an excellent idea. In our study, we instructed the LLM to generate code in a general format, adaptable to any sorting task length. Even when participants included action sequences in their descriptions, the LLM abstracted these sequences into executable and general codes. This abstraction suggests that the LLM can effectively distill the algorithms from the descriptions.
>
> **Concerning positional statements:** As noted, backward algorithms were less accurately identified and predicted. Our supplementary analysis revealed that 6 of 8 standard algorithms exhibit a bias toward the forward version in predictive performance. However, in the recovery test (Fig. A2, appendix), backward algorithms were better identified with LLM self-generated strategy descriptions. We believe that this bias observed in empirical data likely stems from unequal sample sizes in human data and biases in human descriptions.
>
> **On individual differences in participant backgrounds:** Little information is available beyond basic demographics, as participants were recruited online via Amazon’s Mechanical Turk. It is possible that the highest-performing participants had some education in sorting algorithms, providing clearer descriptions. This might also reflect the “good subject effect,” where participants who perform well on one task tend to perform well on all tasks, likely due to their higher investment in the experiment.

---

### Decision · Program_Chairs · 2024-07-10

**Decision:**

Accept

**Comment:**

This paper introduces an approach for using LLMs to analyze the verbal reports that people provide to describe their introspective impression of how they performed cognitive tasks: the authors show that LLMs can be helpful for translating unstructured, introspection-based descriptions into structured code that can be systematically analyzed.

Reviewers pointed out several strengths:
- The proposal is innovative from both the LLM side (it is a novel application of LLMs) and the cognitive science side (it could be helpful in addressing an important question in cognitive science)
- The LLM-generated code is predictive of people’s behavior, suggesting that this approach is capturing something important about human cognition
- The paper is well-written and easy to follow

Limitations noted by reviewers include:
- Only one LLM and task are analyzed; it would enhance the paper to test the generalizability of the approach.
- The utility of this approach to cognitive science is currently mainly speculative. What could really make the paper compelling would be to show some concrete insight that is provided by using LLM-generated code but that would not have been obtained otherwise. That is definitely a lot to ask, and it’s hard to say what this insight would look like, but if it were accomplished it would really strengthen the paper.
- The code that is generated by the LLM reflects to some extent the inductive biases of the LLM (rather than being just determined by the language produced by the participants). However, the authors include some thoughtful discussion of this point - and, more broadly, any approach to interpreting language is guaranteed to rely on inductive biases of whatever does the interpreting, due to the ambiguity of language.

Overall, I believe that this paper is right on the edge - I think it could justifiably be accepted or rejected. Its ideas are interesting and well-executed, but its main utility remains fairly speculative.

[At least one review was discounted during the decision process due to quality]